# ATOM Program System and Computational Experiment

Larissa V. Chernysheva [1,]* and Vadim K. Ivanov [2,]*

1  Sector of Solid State Theory, A.F. Ioffe Physical-Technical Institute RAS, 194021 St. Petersburg, Russia
2  Department of Physics, St. Petersburg Polytechnic University of Peter the Great, 195251 St. Petersburg, Russia
*  Correspondence: larissa.chernysheva@mail.ioffe.ru (L.V.C.); ivanov@physics.spbstu.ru (V.K.I.)

**Abstract:** The article is devoted to a brief description of the ATOM computer program system, designed to study the structure, transition probabilities and cross sections of various processes in multielectron atoms. The theoretical study was based on the concept of a computational experiment, the main provisions of which are discussed in the article. The main approximate methods used in the system of programs for taking many-electron correlations into account and determining their role in photoionization processes, elastic and inelastic electron scattering, the decay of vacancies, and many others are presented. The most significant results obtained with this software are listed.

**Keywords:** atom; RPAE; photoionization; electron-impact ionization; ionization of the inner shells; decay of vacancies; computer





## 1. Introduction

In the theoretical study of multielectron atomic systems, software plays an important role because it is necessary for calculating the energy and spatial structure and characteristics of various processes occurring in atoms. Calculations in the framework of relatively simple one-electron models did not lead to agreement between calculations and experimental data, despite the fact that the pair interaction between electrons is the well-known *Coulomb interaction* [1–7]. In the 1960s–1970s, it became completely clear that in order to describe the structure and reaction of atoms to external influences, it is necessary to go beyond single-electron concepts and take collective multielectron effects into account [3–7]. At the same time, in the theoretical group of the St. Petersburg Ioffe Institute, under the guidance of Prof. Miron Ya. Amusia, began to create a package of applied programs known as ATOM [8–10], which was improved in subsequent years and aided a huge number of direct calculations that were carried out in atomic systems to describe experimental data and predict new many-particle effects. These calculations laid the foundation for the development of the concept of a computational experiment.

Theoretical studies of many-body systems, as a rule, are accompanied by the need to take many-body interactions into account. At the same time, the determination of the spatial and energy structure, as well as the response of such systems to external influences, faces significant computational difficulties, especially in cases where perturbation theory cannot be consistently applied [7–11]. The theory of atoms and atomic particles is also facing such difficulties despite the fact that the pair interaction between electrons has been well-known to physicists for more than a century. Nevertheless, a significant development of methods for describing multielectron atoms and their interactions with external fields began in the second half of the last century when the apparatus of many-body theory was developed, great computational capabilities appeared, and experimental studies of atomic systems began to be widely carried out [8–10]. These experiments showed that the simplest description of atoms based on single-particle models—the Hartree–Fock (HF) approximation, in particular [12]—is unable to describe the characteristics and properties of the behavior of atoms in cases of their interactions with electromagnetic fields and other particles.

One of the most successful theoretical approaches to describing the structure and processes in atoms are approximations based on many-body theory, particularly including perturbation theory [13] and random phase exchange approximation (RPAE) [3,5,6]. It was the RPAE method, developed as the 1960s turned into the 1970s, that first made it possible to describe the photoionization cross sections of multielectron atoms and to show the decisive role of many-particle (collective) interactions in atomic systems. The big advantage of the RPAE is the self-consistency of this approximation, within which the main general laws of the processes of the ionization and excitation of electrons, such as the oscillator sum rule and the condition of gradient invariance, are satisfied [7,8]. Subsequently, the application of this approximation made it possible to describe and explain numerous experimental data and predict many-electron effects that manifest themselves in the processes of the ionization and excitation of atomic systems. It is these calculations that have stimulated numerous theoretical and experimental studies of many-particle processes in various research centers and laboratories.

The theoretical group created by Prof. Miron Ya. Amusia has developed and described the concepts of a computational experiment that is currently considered to be a new methodology and technology of scientific research [14,15]. The implementation of this concept makes it possible to use the capabilities of computers in combination with existing traditional research methods, creating a new style of research work that combines the work of theoreticians, experimenters and computer programmers.

The purpose of this work was to present a brief description of the ATOM program system and its capabilities, as well as a generalization of the results obtained in atomic physics using the developed software. In particular, this paper describes a computational experiment on the development and study using the ATOM system of theoretical models for studying the structure of atoms, simple molecules, endohedral atoms and their interactions with external fields.

## 2. Software for the Computational Experiment

The main task of the computational experiment is to develop numerical models for the specified range of problems within the framework of a pre-selected theoretical approach. The Hartree–Fock (HF) method is taken as the initial approximation, and many-electron correlations in atoms are taken into account in the framework of the RPAE or its simplified modifications [8–10].

The theoretical model of any process under study in the chosen approximation should provide a satisfactory mathematical description of the experiment. First of all, this model includes the choice of the wave functions of atomic electrons in both the ground and excited states, as well as additional external particles participating in the process. The wave functions of the ground state of an atom are determined in the HF approximation. The functions of excited states in the HF method can be found in a self-consistent field or in the field of the "frozen" atomic core for electrons. The choice of the one-particle approximation is determined by both the problem under consideration and the role of many-electron interactions, some of which can be taken into account by the choice of one-particle wave functions. In addition, not only electrons but also other particles can act as additional external particles, e.g., mesons and positrons. The single-particle transition amplitudes are determined in terms of wave functions, which makes it possible to calculate the characteristics of atoms or the probabilities of processes in the HF approximations. Then, the matrix elements of the interaction between electrons in various processes are calculated, and this makes it possible to find the characteristics of atoms or the probabilities of processes while considering multielectron correlations, RPAE in particular [8,9].

The ATOM system was created as a result of many years of research into the structure and processes in multielectron atoms; it was designed to calculate the characteristics of atoms, endohedral fullerenes, and diatomic molecules, as well as the probabilities of the processes of interaction of electrons, photons and other particles, on a computer [3–10,16,17].

Let us briefly list some main characteristics of atoms and processes that are determined within the framework of this system:

- Wave functions of atoms in the ground and excited states in the HF and Hartree–Fock–Dirac approximations.
- Amplitudes and cross sections of photoionization (including the strength of oscillators of discrete transitions) of atoms with filled and half-filled shells while taking the interaction between electrons of one, two or more shells into account.
- Atom polarizability.
- Characteristics of the angular distribution of photoelectrons and secondary electrons both in the dipole and outside the dipole approximation.
- Parameters of the spin polarization of photoelectrons.
- Scattering cross sections of fast electrons through the generalized oscillator strengths, taking the influence of electrons in one, two or more shells into account.
- Angular distribution of secondary electrons arising from the inelastic scattering of fast particles on atoms.
- Phases and cross sections of the elastic and inelastic scattering of particles (electrons, positrons, and mesons) of low and medium energies on atoms.
- Photoabsorption cross sections with allowance for the decay of vacancies and the inelastic scattering of a photoelectron.
- Cross sections of the ionization and excitation of an atom by electron impact.
- Probabilities of the single-electron and double Auger decay of vacancies in atoms.
- Probabilities of the one-photon decay of one- and two-hole states.
- Characteristics of bremsstrahlung of high and intermediate energy incident particles.
- Characteristics of capture of mu-mesons by atoms.
- Amplitudes, photoionization cross sections, and angular anisotropy parameters of endohedral atoms and the decay of vacancies in such atoms.
- Characteristics of the inelastic scattering of fast electrons on endohedral atoms.
- Characteristics of photoabsorption processes in negative and positive ions.

The creation of theoretical models is only the first step in a computational experiment. On their basis, numerical models are constructed and make it possible to obtain an approximate solution of the initial problems on a computer with the required accuracy. The wave functions of an atom in the HF approximation are represented as a product of the radial, angular and spin parts. Radial functions are the solution of the HF equation (or system of equations) through the method of successive approximations. Integration over the angular variables of the wave functions and summation over the spins are carried out analytically and enter the expression for the matrix elements, which are the results of solving integral equations in RPAE [9,10]. The multidimensional integrals in these equations are reduced to one-dimensional ones after separating the corner parts and integrating over the angle variables. The RPAE equations are transformed into a system of algebraic equations, the solutions of which are reduced to the inversion of matrices. The expression for the amplitudes of physical processes, defined in terms of matrix elements in RPAE, often contains summation and integration over intermediate states and may have singularities due to energy denominators [9,10]. The integrals of the emerging singularities are analytically calculated. The resulting expression is "matched" with the contribution of non-singular regions, which is found as a result of numerical integration. In this case, the presence of an imaginary additive in the energy denominator of the calculated amplitudes leads to the fact that the process amplitude has both real and imaginary parts.

The numerical solution of the problem, which makes it possible to study the chosen physical process, is sought without separation from physical considerations, which often lead to model simplification. For example, when discretizing the original model, an important element of the calculation is the choice of a sufficiently high upper limit; up to this limit, instead of integration over the continuous spectrum, summation is carried out on a computer [8,9]. The practice of calculations shows that in the physics of the atom, one has to deal with matrix elements that rather quickly decrease as the energies of the

states entering into them increase. In addition, to improve the accuracy of calculations, it is advisable to choose sampling points so that the largest number is in the region of relatively low energies E. This condition is ensured by using the electron momentum $p = E^{1/2}$ as an integration variable.

### 3. Organization of the Computational Experiment

The optimal organization of the computational process depends on the choice of theoretical and technical characteristics of computational algorithms. The theoretical parameters of the algorithms are related to the formulation of the problem. They include a formal description of the problem, a solution method, the algorithm itself, and its implementation in the chosen language. Technical characteristics depend on the computer used and include system tools, the capabilities and features of the algorithmic languages used, the form of presentation, and the storage of initial information.

The choice of strategy in the development of the algorithm itself includes a number of issues. First, in all possible cases, the complex initial problem is divided into a number of simple ones that are easier to implement on a computer. Due to the allocation of subtasks into independent modules, such as the calculation of dipole and Coulomb matrix elements, integration with a pole, matrix inversion, the problem of using them in the study of new physical processes is simplified. Furthermore, in each of the subtasks, the necessary algebraic simplifications are carried out. In particular, the change of variable required in the calculation of wave functions is carried out in the calculation of all characteristics of atoms, since the matrix elements used in them are expressed in terms of wave functions. Access to auxiliary (intermediate) quantities is as important for a theoretical physicist as measurements are for an experimenter in an experiment. In the processes under study, auxiliary quantities usually refer to the matrix elements that are used to obtain qualitative estimates.

One of the components of the algorithm is the analysis of the accuracy of calculations. The accuracy of theoretical models is generally unknown, and it is established with the help of estimates obtained as a result of roughly approximate calculations. The accuracy of computer calculations can be very high. It is advisable to use an accuracy that it is somewhat higher than the expected error of the physical result itself. Often the error of intermediate results is more important, since the value obtained in a physical experiment is found as the difference of large numbers. In addition, in the process of computing, the error can accumulate, and, as a rule, it is difficult to evaluate it. Therefore, the final result on a computer often does not require a level of accuracy as high as that needed for intermediate data and that realized at intermediate stages. All this is considered when choosing numerical methods for solving problems.

The development of computational algorithms is the second stage of the computational experiment. In the next steps, computer programs are compiled that implement the selected algorithms. The ATOM system uses numerical methods for the theoretical study of the structure of complex atoms and the processes occurring with their participation [11]. The high-level language Fortran, which is the main language in many physical calculations, is used to write the algorithms. To facilitate the implementation of the ATOM system on other computers, Fortran does not use the features and extensions of languages implemented in some translators. Algorithms are recorded in accordance with the respective technology. Algorithms are designed in the form of modules, which are divided into three groups. Modules of the first type are procedures without formal parameters and include the description of variables, the input and printing of initial data, the description of the algorithm containing printing of intermediate values, and the output of results. These are executive programs, each of which solves an independent physical problem. Modules of the second type (specialized) are programs or function procedures with formal parameters. They contain descriptions of the variables and the algorithm. Modules of the third type (service) contain descriptions of variables, the input and printing of initial data, or the printing of results. These modules are used in the development and assembly of modules of the first type. Modules of the fourth type (generic) implement standard mathematical

methods. For clarity and ease of finding errors, variables in modules are grouped according to their purpose; variables common to all modules are assigned permanent identifiers, and the names of the variables are usually abbreviated names of the physical processes under study. The input of initial data is accompanied by the printing of all physical quantities. The same is done for intermediate values and program results. Detail printing turns a software module into a theoretical physicist's tool; it plays the same role as diagnostics in a natural experiment for an experimenter.

The ATOM system includes an application program (AP) and a database (DB). The AP contains more than 50 executive modules (according to the number of physical tasks to be solved), more than 10 service modules, more than 70 specialized modules and more than 16 generic modules, and it has a hierarchical structure. During its development, the following requirements were taken into account: the modular principle of organization, the constant expansion of the system's capabilities, the convenience of users, ease of implementation, and the use of the system as part of the software for the computing network for collective use. The ATOM system was built with basic tools, so it is easily implemented and widely used in many places. For each atom and each physical process under study, the DB contains the required wave functions, input and output physical characteristics.

The input language of the ATOM system belongs to the class of task languages that allow for a wide class of users who do not have special training in programming to work.

An important stage of the computational experiment is the implementation of computer calculations, during which the capabilities of the created programs are constantly expanding. At present, the ATOM system enables the solving of the following problems [8,9]. In the Hartree and HF approximations, one can obtain various wave functions, namely wave functions of the ground state of an atom, wave functions of excited states consistent with the functions of the ground state, wave functions of excited states in a continuous spectrum for given energies in a fixed field of an atom with or without orthogonalization to the wave functions of the ground state, wave functions of excited states in the discrete spectrum for given values of the principal quantum number in a fixed core field with or without orthogonalization to the wave functions of the ground state, and wave functions of the mu-meson and positron.

Because it has a set of necessary wave functions in the DB, the ATOM system allows one to determine, in the HF and RPAE approximations, matrix elements within one or two transitions, each of which is characterized by one wave function of the ground (hole) state and a set of wave functions of excited (partial) state discrete and continuous spectra for a finite series of energies, namely dipole matrix elements of the length and velocity form, the Coulomb matrix of effective interaction, matrix elements of the terms of the expansion of a plane wave in a series in terms of Legendre polynomials, and Coulomb matrix elements such as "three particles–one hole", "two particles–two holes", and "three holes–one particle". The resulting matrix elements are the basis for studying a number of processes.

Using the example of calculations of the photoionization processes in the framework of the RPAE, we present the main stages of calculations. First, the wave functions of the ground and excited states are determined in the HF approximation. Using the obtained wave functions, the dipole matrix elements of phototransitions to discrete states and states of the continuous spectrum are calculated in the zeroth approximation. In the next stage, the matrix elements of the Coulomb interaction between all the HF states involved in the process, both real and virtual, are calculated. Furthermore, integral equations are solved in the program to determine the amplitudes and cross sections of phototransitions while taking multielectron correlations within the framework of the RPAE into account.

Thus, the analytical expression for the photoionization amplitude upon the absorption of a quantum with frequency ω, considering intra- and intershell interactions, can be represented in the following symbolic form [9,11]:

$$\langle v|\hat{D}(\omega)|i\rangle = \langle v|\hat{d}|i\rangle + \left( \sum_{\substack{k_2 > F \\ k_1 < F}} - \sum_{\substack{k_1 > F \\ k_2 < F}} \right) \frac{\langle k_2|\hat{D}(\omega)|k_1\rangle\langle v,k_1|\hat{U}|i,k_2\rangle}{\omega - E_{k_2} + E_{k_1} + i\delta(1 - 2n_{k_2})} \quad (1)$$

Here, $\langle v|\hat{d}|i\rangle$ and $\langle v|\hat{D}(\omega)|i\rangle$ are the dipole matrix elements of the transition between the $|i\rangle$-initial and $\langle v|$-final states in the HF and RPAE approximations, respectively. The matrix element of the interaction between electrons involved in the transition between intermediate $|k_1\rangle$ and final $|k_2\rangle$ states is determined by the sum of the direct and exchange matrix elements.

$$\langle v,k_1|\hat{U}|i,k_2\rangle = \langle v,k_1|\hat{V}|i,k_2\rangle - \langle v,k_1|\hat{V}|k_2,i\rangle \quad (2)$$

where $\langle v,k_1|\hat{V}|i,k_2\rangle$ is the matrix element of the Coulomb interaction. Summation (integration) over the intermediate states $|k_1\rangle$ and $|k_2\rangle$ with energies $E_{k_1}$ and $E_{k_2}$ involved in the process above and below Fermi level $F$ is carried out in both a time-direct process and a time-reversible process; $n_k$ is the Fermi step $n_k = \begin{cases} 0\ k > F \\ 1\ k < F \end{cases}$.

When only intrashell correlations are taken into account, summation in (1) is carried out only over intermediate excited states of electrons in one shell under consideration. When intershell interactions are taken into account in the summation, the correlation term takes the transitions of electrons in different shells involved in the process into account.

As a result of Solution (1), we obtain the transition amplitude in the RPAE approximation, which is substituted into the formulas for determining the partial or total photoionization cross sections ($\hbar = m_e = e = 1$) [11]:

$$\sigma_{i \to v}(\omega) = \frac{4\pi^2}{\omega c} \int |D_{iv}(\omega)|_2 \delta(E_v - E_i - \omega) dv \quad (3)$$

where $\omega_{vi} = E_v - E_i$ is the transition energy and $v$ is the total set of quantum numbers that characterize the final state.

## 4. Interference Effects in the Processes of Photoionization of Atoms

The photoionization cross section (oscillator strengths) and the anisotropy coefficients of the angular distribution of photoelectrons are expressed in terms of the dipole matrix elements of the coordinate or momentum and are determined by considering the interactions of all electrons of the shell under study with each other, as well as intershell and intersubshell interactions.

Within the framework of the one-particle HF approximation, it was not possible to obtain agreement between the theory and the available experimental data, on photoabsorption in atoms in particular. In addition, the results of calculations in the HF approximation did not obey the conditions of gauge invariance, since the photoionization cross sections obtained with different transition dipole operators gave different results. It became obvious that it is necessary to go beyond the HF approximation when describing photoionization processes. This means, by definition, that it is necessary to consider the many-electron correlations caused by the part of the electron–electron interaction that is neglected in determining the self-consistent mean field.

The correlation interaction between the electrons of an atom can be taken into account using a number of theoretical methods (review [18]). Those that use the apparatus of the theory of many bodies and apply the diagram technique, namely RPAE [5,11,19] and many-particle perturbation theory (MPT) [20], are widespread. Based on the HF approximation as

the zeroth approximation, the apparatus of many-body theory makes it possible to represent the mechanism of any process under consideration in the lowest nonvanishing order of perturbation theory in terms of interelectronic interaction and to present corrections to it in higher orders.

The development of the RPAE to account for many-electron correlations in atoms was the next step after the HF approximation in the creation of self-consistent approximations in many-body theory. From a many-body perspective, RPAE takes an infinite sequence of perturbation series terms of a certain class (class of diagrams) that contribute the most in each order of perturbation theory into account. The self-consistency of the approximation and its advantage also lie in the fact that, within its framework, one can obtain the gauge invariance of the obtained photoionization cross sections, namely the equality of the results in calculations with different types of dipole operators (in contrast to the HF approximation).

The first natural step in considering the correlation interaction, sometimes called residual, was to consider it between the electrons of only one shell, since the latter are well-separated spatially and energetically from the electrons of other shells. In application to the study of the processes of the ionization and excitation of atoms, correlations were first successfully taken into account in the framework of the RPAE [5,11]. In these calculations, it turned out to be sufficient to take the residual interaction between the electrons of the ionizable shell (intrashell interaction) into account in order to obtain satisfactory agreement with the experimental data on the total photoionization cross sections in the photon energy range from ionization thresholds to several hundreds of electron volts. It turned out that the residual interaction between electrons is essential for all outer and intermediate multielectron shells ($p^6$, $d^{10}$, and $f^{14}$), which make the largest contribution to the total photoionization cross section.

Intrashell correlations can significantly change the value of the photoionization cross section but usually do not lead to its qualitative changes depending on the energy. Intershell interaction, on the contrary, often leads to qualitative changes.

In partial ionization cross sections of few-electron shells whose electrons participate in relatively "weak" transitions, the role of intrashell correlations is usually small. However, when describing the partial cross sections for the ionization and excitation of such electrons, it is necessary to take intershell correlations (intershell effects) into account, since few-electron shells, e.g., $ns^2$, are subject to a strong screening effect of the surrounding many-electron shells. In other words, their behavior is completely collectivized and determined by the surrounding many-electron shells. Although the atomic shells are well-separated from each other spatially and energetically, considering the connection of electrons in different shells turns out to be very important in describing a number of dynamic processes in the atom. Intershell interactions manifest themselves most strongly in the ionization cross sections of few-electron shells, total ionization cross sections at the thresholds of inner shells, the angular distribution and polarization of photoelectrons, the decay of vacancies in inner shells, and photoelectron spectra. The prediction of a significant effect of many-electron shells on few-electron shells [7,11,21], the complete loss of their individuality by the latter, and their consequent collectivization served as the impetus for a wide experimental and theoretical study of the manifestations of intershell interactions.

Intershell interactions most clearly manifest themselves in the study of the photoionization cross sections of $5s^2$ shells in atoms whose electrons have completely lost the features of individual behavior. In this case, it is appropriate to speak of the collectivization of $5s^2$ electrons under the influence of the surrounding $4d^{10}$ and $5p^6$ multielectron shells. Under the influence of external electrons, a minimum appears in the partial cross section for the photoionization of $5s$ electrons, followed by a maximum under the influence of electrons from the inner shell. Such collectivization is typical for the $5s^2$ shells of many elements, beginning with Cd (Z = 48). However, the dependence of the photoionization cross section on energy undergoes successive changes with increasing nuclear charge Z. Thus, when going from Xe to La, the interference minimum shifts to the region of the

discrete excitation spectrum. It should be noted that the collectivization of $ns^2$ electrons was initially theoretically predicted and then confirmed by experimental measurements.

The study of the angular distribution and polarization of photoelectrons in principle provides more detailed information about dipole transitions in atoms and the effect of many-electron correlations, since the parameters describing these processes are determined by the transition amplitudes together with the scattering phases of a photoelectron in the ion field. Thus, the angular distribution of photoelectrons knocked out of a shell with quantum numbers *n, l* when an unpolarized atom is irradiated with unpolarized light is determined by the expression [7–9]:

$$\frac{d\sigma_{nl}}{d\Omega} = \frac{\sigma_{nl}(\omega)}{4\pi}\left[1 - \frac{1}{2}\beta_{nl}(\omega)P_2(\cos\theta)\right] \qquad (4)$$

where $P_2(\cos\theta)$ is the Legendre polynomial, $d\Omega$ is the element of the solid angle of emission of a photoelectron, and $\sigma_{nl}(\omega)$ is the total photoionization cross section of the *nl*-shell (obtained from (1)–(3)). The angular anisotropy parameter $\beta_{nl}(\omega)$ is expressed in terms of dipole matrix elements and photoelectron scattering phases with angular momenta ($l \pm 1$).

The intrashell interaction, as a rule, does not lead to qualitative changes in the dependence of the anisotropy parameter in comparison to a single-particle calculation. The situation is different when the intershell interaction is taken into account: the transition amplitude can acquire additional zeros, maxima, and minima, which is reflected in quantitative and qualitative changes in the partial photoionization cross sections and angular distributions of photoelectrons [7,11]. A striking example is the behavior of the angular anisotropy parameter of electrons in the $5p^6$ shell [22,23].

The study of the polarization of photoelectrons also makes it possible to obtain even more detailed information about the behavior of dipole amplitude. The experimental determination of partial cross sections, angular distributions of photoelectrons, and their polarization forms a so-called full quantum-mechanical experiment, which makes it possible to measure all the amplitudes characterizing photoionization, including their real and imaginary parts. Calculations have shown that the degree of polarization of photoelectrons as a function of energy is very sensitive to variations in the amplitudes of dipole transitions and, consequently, to manifestations of intershell interactions [11,24].

Many-electron effects are often more significant in negative atomic ions [11,25] than in neutral atoms, since the interaction between outer electrons is relatively stronger due to more complete screening of the Coulomb field of the nucleus. The formation of negative ions mainly occurs due to the polarization attractive interaction between an additional electron and a neutral atom. Therefore, in determining the cross sections for the photodetachment of electrons from negative ions, in addition to RPAE correlations, it is necessary to take the polarization potential into account. The latter can be conveniently achieved by redefining the wave functions of the additional electron: instead of the HF functions, use Dyson orbitals, which take the self-energy parts of the external electron into account [25].

Let us list the most important results obtained in the study of many-electron correlations in the processes of photoionization and photoexcitation. Most of the results were obtained using the ATOM system [7–11,16,17,24].

1.  It has been demonstrated that giant resonances in the photoionization cross sections of a number of atoms, xenon (Xe) and its neighbors such as iodine (I), cesium (Cs) and lanthanides in particular, are completely multielectron in nature. They are analogues of plasmons in solids, and at least all ten electrons of the $4d^{10}$ subshell participate in their formation in atoms.
2.  It was predicted that the action of multielectron shells qualitatively changes the photoionization cross section of few-electron subshells, leading to a new continuous spectrum, the so-called *interference resonances* [6,11]. These resonances are a direct

consequence of the interaction between electrons that belong to different subshells or even shells, and they have indeed been observed in many atoms.

3. It has been shown that the interelectronic interaction and the correlation effects caused by it are sharply manifested not only in the photoionization cross sections but also in the angular distributions of photoelectrons [22] and spin orientation [24].

4. It has been demonstrated that multielectron correlations significantly affect the photoionization cross sections of outer and intermediate atomic subshells in a very wide frequency range. These effects are especially strong near the subshell ionization thresholds, but they are quite noticeable at relatively high photon frequencies of up to 1.5 or 2 keV and even above the threshold values [7,9,11].

5. Researchers have identified specific examples of the strong interaction of electrons belonging to two or even three different subshells, which leads to strong changes in their cross sections far from any ionization threshold, including the formation of completely new maxima [21,26].

6. It has been shown that along with the main line, which corresponds to the removal of an electron from a given subshell, there are satellite and shadow lines of a pure many-electron nature in the photoelectron spectrum. They appear at any incoming frequency of photons with the same strength relative to the main line. Quite often, the strength of these lines is high, which is a direct manifestation of the very important role played by the interelectronic interaction [27].

7. It has been demonstrated that the interelectronic interaction can in some cases be so strong that single-electron lines completely disappear [27] This effect, called the "melting" of electron shells, has been observed in Xe and some of its neighbors. In atomic physics, this effect is less common than other manifestations of correlations, but it is of great importance as an example of the possible power of the interelectronic interaction.

8. It has been shown that one photon, which can interact with only one electron, is able to simultaneously remove two or even more electrons from an atom. This process only occurs due to the presence of interelectronic interactions when the electron of the inner shell is removed from the atom and is accompanied by the Auger effect [7,9,16]. One photon can simultaneously remove two or even more electrons from the outer subshell, even if the photon frequency is below the intermediate photoionization threshold.

9. The autoionization of the continuous spectrum was predicted [25,28] due to the existence of relatively narrow resonances in the photoabsorption cross section of negative ions that arise as a result of the strong interaction of electrons belonging to two outer subshells.

10. Researchers have identified strong autoionization resonances that arise due to the effective interaction between the discrete excitation "two electrons–two vacancies" from one subshell with the continuum "one electron–one vacancy" of the other [29]. Therefore, almost everywhere, with the exception of the immediate vicinity of the first ionization potential, the photoionization cross section has a fine structure consisting of narrow autoionization resonances.

11. Researchers have predicted unexpectedly large low-energy non-dipole corrections in the angular anisotropy of photoelectrons, which lead to the creation of resistance currents that are quite observable macroscopic joint effects during the photoionization of atomic gases [30].

The intershell interactions in atoms with open *np*- and *nd*-shells have been theoretically and experimentally studied to a lesser extent than in atoms with filled shells. The reasons for this are not only computational difficulties but also experimental ones associated with obtaining these atoms in a vapor state. There are a number of calculations of the photoionization cross sections for atoms of groups VI and VII of the periodic system in the single-particle approximation while taking correlations into account. These calculations indicate that the correlations between electrons—both intrashell and intershell—should

be no less in these atoms, and sometimes even more, than in atoms neighboring in the periodic system with filled shells.

## 5. Many-Electron Effects in Electron-Impact Ionization Processes

The study of the ionization of atoms by fast electrons (or other particles) makes it possible to trace the dependence of intershell interactions on the momentum $q$ and angular momentum $\Delta l$ transferred during scattering. The transfer of various moments to an atom leads to (along with dipole moments) monopole, quadrupole, and other transitions in the atom, thus making it possible to find out the role of the intershell interaction components of different multipolarities. The cross section for inelastic scattering of fast electrons on atoms is determined in terms of the generalized oscillator strengths (GOS), which are calculated for transitions of different multipolarities and describe the reaction of an atom to the transfer of momentum and energy to it. The differential cross section for inelastic scattering of fast electrons is proportional to the density of the GOS.

In the limiting case of the transferred momentum, only the dipole component of the interaction "survives"; therefore, in ionization by fast electrons, the influence of intershell correlations manifests itself similarly to that in the process of photoionization. Following the change in the influence of the intershell interaction with increasing momentum transfer $q$ using the dipole-density component GOS as an example, we note that with increasing $q$, the influence of the outer shells on the ionization of the deeper ones decreases. The reason for this is that as $q$ increases, the incident electron penetrates deeper and deeper into the atom and the effective radius $r$ of the interaction with it finally becomes smaller than the radius of the outer shell. The outer electrons shield the inner shell from the impact of the incident electron to a lesser extent [11,16].

The differential cross section for inelastic scattering of fast electrons is proportional to the density of the GOS $\partial f(\omega, q)/\partial \omega$ and can be written as [31]

$$\frac{d^2\sigma}{d\omega \, d\Omega} = \frac{4\pi}{\omega E} \frac{\partial f(\omega, q)}{\partial \omega} \frac{d \ln q^2}{d\Omega} \tag{5}$$

where $E$ is the energy of the incident electron; $\omega$ and $q$ are the energy and momentum, respectively, transferred to the atom during scattering; and $d\Omega = 2\pi \sin\theta d\theta$ is the element of the solid angle into which the incident electron was scattered. When many-electron correlations are taken into account, instead of the single-electron matrix element included in (5), the corresponding matrix element—which is determined by expressions similar to those written for the dipole component of the photoionization amplitude—is substituted.

The differential cross section is proportional to the total density GOS. For small transferred momenta $q$, the GOS is determined by the contribution of the dipole component. As $q$ increases, the contribution of monopole and especially quadrupole transitions increases, for which the influence of surrounding shells is smaller than for dipole ones.

As $q$ increases, the influence of the outer shells on the ionization of the deeper ones decreases. The aforementioned reason for this is that as $q$ increases, the incident electron penetrates deeper and deeper into the atom and the effective radius $r_{\text{э}}$ of the interaction with it becomes smaller than the radius of the outer shell. The outer electrons shield the inner shell from the action of the incident electron to a lesser extent [32].

On the contrary, the influence of inner shells on the ionization of electrons from outer shells can remain significant even at sufficiently large transferred momenta $q$. Moreover, since the rate of decrease in the ionization amplitude is determined by the product $qr_{\text{э}}$ and the radius of the inner shell is less than the radius of the outer one, the contribution of the direct amplitude usually decreases with increasing $q$ faster than the correlation one. Thus, the relative role of internal electrons can increase.

As in the process of photoionization, the scattering cross section for fast electrons with ionization of outer s-electrons has a collective character. The influence of the surrounding multielectron shells on their ionization leads to qualitative changes in the dependence of the cross section on the transferred energy and momentum [11,32].

When studying the scattering of slow electrons by atoms, information about the role of many-electron correlations and the probability of the process can be obtained by determining the self-energy part of a hole or a particle of a single-particle Green's function in a simplified RPAE.

The cross section for elastic scattering of electrons with energy $E$ is expressed in terms of the scattering phases $\delta_l(E)$ of partial waves with moment $l$. Using the ATOM system, RPAE corrections $\Delta\delta_l(E)$ to the HF phases of elastic scattering $\delta_l^{HF}(E)$ are calculated [9,11].

$$e^{i\Delta\delta_l(E)} \sin \Delta\delta_l(E) = -\pi \langle El | \hat{\Sigma}(E) | El \rangle \tag{6}$$

where $\langle El | \hat{\Sigma}(E) | El \rangle$ is the matrix element of the polarization interaction of the incident electron with the atom. The self-energy part of the Green's function depends on the energy of the incident electron and describes the nonlocal interaction between the incident electron and the electrons of the atom. This approach first made it possible to describe experimental data on the elastic scattering of slow electrons by a significant number of atoms with high accuracy and without using the phenomenological polarization potential. The total cross section of inelastic scattering is expressed in terms of the imaginary part of the phase shifts (6) $\text{Im}\delta_l(E)$.

The same methods were applied to the description of the elastic scattering of slow positrons by atoms [11,33]. In contrast to the scattering of electrons, in this problem, on the one hand, it is not necessary to consider the exchange interaction, but it was necessary to consider the formation of a bound state, such as positronium, which complicates its solution.

Below, we present the most important results of the study of electron or positron scattering, both elastic and inelastic [8–11,16]. Some of the obtained results are also important for understanding the collisions of atoms involving heavy charged particles, such as protons and $\mu$-mesons.

1.  It was demonstrated that in the cross section of the elastic scattering of electrons on atoms, there is a Ramsauer minimum that arises due to the action on the incoming electron, along with the self-consistent HF potential and the polarization potential. This potential has a purely many-electron nature. This leads to Ramsauer minima in the electron scattering cross sections, not only on atoms of the noble gases Ar, Kr, and Xe but also on alkaline earth elements such as Ca. In the latter atoms, the Ramsauer minima have very high energies [11,25,34].

2.  The polarization potential turned out to be strong enough to form stable negative ions of a number of atoms with filled subshells, alkaline earth elements in particular [25,34,35], although with a very low binding energy. These negative ions have been observed in experiments.

3.  The important role of the polarization potential in the elastic and inelastic scattering of slow positrons by atoms was investigated and discovered. It has been demonstrated that the possibility of an incoming positron to form a bound state (called positronium) with an external atomic electron during scattering, greatly affects polarization potential [11,36]. Accounting for this temporary formation of positronium can lead to bound states of a positron with an atom, which is a completely new kind of positively charged ion [37].

4.  The temporary formation of positronium during the elastic scattering of positron atoms can, in principle, lead to a qualitative modification of the polarization potential, i.e., it can become repulsive instead of always being attractive [38]. This reversal of the sign of the potential explains the qualitative difference between the low-energy elastic scattering cross sections of a positron on He and a positron on Li and also why the former is orders of magnitude smaller than the latter.

5.  It has been demonstrated that many-electron correlations play a very important role in inelastic collisions for practically any projectile energy. The cross section of inelastic scattering is strongly affected by not only such collective excitations as dipole giant resonances but also multielectron nondipole excitations [10,11].

6.      It was shown that for incident particles with energies close to the excitation thresholds of the intermediate or inner shell, the spectra of the inelastically scattered projectile are strongly modified due to the Auger decay of the created vacancy. This effect is called the post-collision interaction [39]. The many-particle theory of this effect can be found in [11,40].

7.      A new mechanism for the generation of continuous spectrum electromagnetic radiation in inelastic collisions in atoms was proposed. This radiation, called atomic or polarization bremsstrahlung, is mainly due to the dipole deformation of the target atom during the collision. This radiation is strongly influenced by the interaction between atomic electrons and the collective effects caused by it [41].

## 6. Ionization of the Inner Shells of the Atom and Decay of Vacancies

Collective effects during the photoionization of inner shells near their threshold are more complex than those of outer shells. Along with the forced joint ordered motion of electrons of one or several neighboring shells occurring under the action of an external electromagnetic field, various relaxation processes also take place. Relaxation, or rearrangement, is a complex dynamic process that reflects the reaction of atomic electrons to the appearance of a vacancy in one of the shells and its subsequent decay. Accounting for the rearrangement leads to changes in the photoionization amplitudes and the interaction between electrons.

The simplest method that considers the reaction of atomic electrons to a vacancy appearing in an atom after photoabsorption is the "static rearrangement" approximation. In this case, during the decay time of the hole, the photoelectron with low energy does not have time to move far enough from the remaining ion and almost immediately moves into the field of the ion field that has already been rearranged due to decay [7,9].

The "static" rearrangement approximation becomes inapplicable when the lifetime of a vacancy $T_{nl}$ in a shell with quantum numbers $nl$ is comparable to the photoelectron escape time $t$ from an atom. In this case, it is necessary to take the dynamics of the process into account. During Auger decay, the new field acting on a photoelectron corresponds to a charge one greater than the initial field formed during the absorption of a quantum. As a result of increased attraction, the slow electron in the new field has an energy lower than that which it would have if the decay of the vacancy was neglected. The released energy is carried away by a fast Auger electron. This phenomenon, associated with the redistribution of energy between a photoelectron and an Auger electron, is a strong correlation effect called the post-collision interaction (PCI) [39,40]. This effect significantly changes the amplitude and, accordingly, the photoionization cross section, as well as the energy distribution of Auger and photoelectrons. Recently, the effects of PCI have been intensively studied both experimentally and theoretically. An analytical study of the amplitude of the photoprocess, taking the PCI into account, showed a redistribution of energy between electrons: a fast electron is accelerated while a slow one is decelerated [40]. The analysis of this correlation effect is simplified when the width of the deep vacancy is not too large and (as a result of its decay) a sufficiently fast electron is formed, so that the interaction with a slow photoelectron can be neglected.

Many-electron correlations are clearly manifested in the decay of vacancies formed during the interaction of photons, electrons, or positrons with atoms. Decay with the emission of even one electron (Auger decay) is in itself a manifestation of the interelectronic interaction. However, in some decay processes, the role of interaction with other electrons of the atom (many-electron correlations), which are not directly involved in the decay, is especially important.

Through the matrix element of the energy proper part of the single-particle Green's function, taken between the wave functions of the occupied states, the shift of the ionization potential is expressed in comparison with its HF value due to the correlation interaction of atomic electrons. The imaginary part of this matrix element gives the total width of the hole level with respect to the Auger decay. In addition to direct Auger decay, in which another

atomic electron is removed as a result of the transition of an electron from an occupied level to a free one, a more complex process (which proceeds due to the effective interaction of electrons and is taken into account in RPAE) is also investigated.

The main results obtained in the study of the decay of vacancies are presented below.

1.  Taking the interaction between the electrons of an atom into account, it was shown that the probability of both radiative and non-radiative—i.e., flowing with the emission of electrons (Auger electrons)—decays can be significantly reduced or increased. Owing to correlations, radiative decay can be completely blocked, which is called the *radiative self-blocking of vacancies* [11].

2.  It has been demonstrated that a state with one vacancy can decay upon the simultaneous emission of one electron and one photon, several electrons, or several photons [42]. At the same time, states with two vacancies can decay upon the emission of one photon or electron [16,17].

3.  It was shown that the energy of an Auger electron leaving an atom after the decay of an internal vacancy shifts towards higher energies when a vacancy is created near the threshold of its formation. This shift increases with a decrease in the energy of the photoelectron that leaves the atom when an internal vacancy is created. This shift is a manifestation of the interaction after the collision [11,16].

4.  The increase in the energy of the Auger electron mentioned in the previous paragraph can be so large that a slow photo- or inelastically scattered electron does not actually have enough energy to leave and remains in the atom because it is intercepted by one of the higher excited levels of the residual ion.

5.  The creation of the innermost shell vacancies leads to avalanches of secondary electrons. Most of them are formed as a result of multistage Auger decay. However, multielectron Auger processes also play an unexpectedly large role in the creation of these avalanches.

## 7. Conclusions

With the help of the developed ATOM software, a huge number of calculations of various atomic processes have been carried out, in most of which multielectron processes play a decisive role. These calculations can be considered the result of a computational experiment in which many-particle effects are studied for a well-defined and well-known pair interaction between electrons. Naturally, in this short review of the obtained results and the possibilities of applying the presented experiment, not all processes associated with the reaction of the electronic system of atoms to external influences are discussed.

In the processes mentioned above, relativistic effects that have little impact on the partial photoionization cross sections of individual shells were neglected. However, when describing photoprocesses in atoms, such characteristics that are directly related to relativistic corrections to the interaction are also determined. One of them is the parameter called the *"branching ratio"* [18], which characterizes the relative probability of the ionization of shell sublevels with different total angular momenta j. The presence of a spin in an electron leads to the fact that during the photoionization of shells, the remaining ion can be in states that differ in the total momentum $j = l \pm 1/2$.

In the process of the photoionization of internal atomic shells with a total moment $j > 1/2$ ($l > 0$), the resulting ions have a certain alignment (the predominant orientation of the total moment $j$) along the direction of the incident photon beam. This alignment, which arises as a result of the uneven population of states with different projections of the total momentum of the ion $M$, manifests itself in the anisotropy of the angular distribution of emission photons or Auger electrons emitted during the decay of a vacancy. The alignment of ions depends on the squares of the dipole amplitudes in a different way than the photoionization cross section, and the measurement of the angular anisotropy of electrons or photons therefore also provides additional independent information about the photoionization process.

The final stage of the computational experiment is the analysis of the results and their comparison with the experimental data. The ATOM system has turned out to be very effective in solving a wide class of problems in the study of atomic processes, including processes involving atomic ions, endohedral fullerenes, diatomic molecules, and atomic clusters, as well as processes in a number of problems in solid state physics.

The ATOM system was created on the initiative and with the active participation of Prof. Miron Ya. Amusia. It was his ideas that led to the creation of the self-consistent RPAE approximation, which was the next step in many-body theory after the HF approximation. The development of the proposed approach and the creation of the ATOM complex of computational programs continued for more than a year of research in the field of physics of multielectron atoms. The development of such systems will make it possible to carry out mass molecular calculations, as well as calculations of atoms placed in strong electric or magnetic fields. The obtained results will become the theoretical base, the starting point for comparing theory and experiment. Thus, the area of applicability of the approach that a group of theoretical physicists under the leadership of Miron Ya. Amusia developed in the theory of the atom will be significantly expanded, providing experimenters with theoretical results of initial approximations that have a relatively high accuracy "on average".

The results of a computational experiment using the ATOM system can be found in supplementary materials and in a number of our monographs [7–11,16,17,19,43].

**Supplementary Materials:** The following supporting information can be downloaded at https://www.mdpi.com/article/10.3390/atoms10020052/s1. List of main publications in which the results were obtained using the ATOM program system.

**Author Contributions:** Conceptualization, L.V.C. and V.K.I.; methodology, V.K.I.; software, L.V.C.; writing—original draft preparation, L.V.C. and V.K.I.; writing—review and editing, L.V.C. and V.K.I. All authors have read and agreed to the published version of the manuscript.

**Funding:** This research received no external funding.

**Data Availability Statement:** Not applicable.

**Acknowledgments:** We consider it our duty to thank Kheifetz A. for his advice and help in preparing our manuscript for publication.

**Conflicts of Interest:** The authors declare no conflict of interest.

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
