# Peer review of "ATOM Program System and Computational Experiment"

_atoms, doi:10.3390/atoms10020052_

Round 1

Reviewer 1 Report

The manuscript reports a description/review of the computer program ATOM, applied in the study of multielectron atoms. In general, the manuscript is well written and clearly explained and is the in scope of the journal. Thus, it should be accepted for publication.

* I have noted some small issues in the pdf file. In particular, there are some typo issues in eqs (1) and (2) and in line 277

Reviewer 2 Report

This review is devoted to the ATOM computer program system, used to study the structure of multielectron atoms, transition probabilities, and cross-sections of various processes. The authors presented a brief description of the main programs entering the package created over 50 years ago and developed to the present day. They also listed the main results obtained in the framework of this approach. The work contains an extensive list of references, where an interested reader can get acquainted with the use of the method in more detail. I recommend the article for publication in the journal.

I have a few minor remarks that the authors could pay attention to clarify certain points.

1) line 140: corner parts --> angular parts

2) line 173: By "dipole matrix elements", do the authors mean the matrix elements of the electric dipole operator?

3) line 265, Eq.(1): The quantity E_F should be determined.

4) line 301: By "different transition dipole operators" do the authors mean the length and velocity gauges of the electric dipole operator?

5) line 324: "called residual". In my understanding, the authors say about "residual Coulomb interaction".  If yes, it might be reasonable to specify it explicitly.

6) line 351: Which shells are related to "many-electron" and "few-electron" shells. Is it possible to specify the largest number of electrons that can be located on a "few-electron" shell?

7) line 386: What is "transition amplitude" in this context?

8) line 662:  "can be considered the result" --> " can be considered as the result"

9) Lines 672-674: It is difficult to understand this sentence. Maybe rephrase it ?!

Reviewer 3 Report

Manuscript ID: atoms-1707965
Type of manuscript: Review
Title: ATOM program system and computational experiment
Authors: L. V. Chernysheva, V. K. Ivanov

This review article describes the computer program system
named ATOM and discusses its applicability to the various
fields in atomic systems in view of the computational experiment.
The authors placed the stress especially on the effect of 
many electron correlations that can be discussed as beyond the 
HF approximations. 
The paper is well organized and well informative as a review 
of the achievement of the program ATOM of which construction 
has been conducted by Prof. Miron Ya. Amusia.
The paper may deserve for publication as an review article in 
an MDPI journal atoms.
However, there are in the article no tables or figures that may visually 
help the readers to understand the authors' ideas to explain.
This makes the article very difficult to understand the description.
The authors would be strongly suggested to include in the paper,
for example, a block diagram that shows the calculation flow of 
the ATOM program system, list of the areas of applicability of the 
ATOM program system, and others.
I am not sure if an article without any tables or figures 
can be advisable to be published as one that matches the
criterion of the MDPI journal atoms. 
I suppose that belongs to the decision of the editorial policy.

Typo: 

line 8: approximate methods should read approximation methods
line 29: of which a should read of a.
line 30: remove were carried out.
line 141: into a system should read into a set.
line 142: the solution of which should read of which solution.
line 206: execuive should read executable.
line 277: a russian letter between E_k1 and E_k2 should be replaced by and.
line 344: collectivized may better be collectiveness throughout the paper.
lines 488-491 and lines 507-511, duplicates. One of them should be removed.
line 527: moment should read angular momentum.
line 540: shifts (6) Im... should read shifts Im...
line 615: intensively should read extensively.

Round 2

Reviewer 3 Report

Al the points I pointed out have been considered appropriately by the authors, and the manuscript has been revised accordingly.

I recommend the paper be published in a present form.

Author Response

Thank you!